# Role of chemisorbing species in growth at liquid metal-electrolyte interfaces revealed by in situ X-ray scattering

Andrea Sartori [1,4], Rajendra P. Giri [1], Hiromasa Fujii[1,5], Svenja C. Hövelmann [1,2], Jonas E. Warias[1], Philipp Jordt[1], Chen Shen [2], Bridget M. Murphy [1,3] ✉ & Olaf M. Magnussen[1,3] ✉

Liquid-liquid interfaces offer intriguing possibilities for nanomaterials growth. Here, fundamental interface-related mechanisms that control the growth behavior in these systems are studied for Pb halide formation at the interface between NaX + PbX$_2$ (X = F, Cl, Br) and liquid Hg electrodes using in situ X-ray scattering and complementary electrochemical and microscopy measurements. These studies reveal a decisive role of the halide species in nucleation and growth of these compounds. In Cl- and Br-containing solution, deposition starts by rapid formation of well-defined ultrathin (∼7 Å) precursor adlayers, which provide a structural template for the subsequent quasi-epitaxial growth of c-axis oriented Pb(OH)X bulk crystals. In contrast, growth in F-containing solution proceeds by slow formation of a more disordered deposit, resulting in random bulk crystal orientations on the Hg surface. These differences can be assigned to the interface chemistry, specifically halide chemisorption, which steers the formation of these highly textured deposits at the liquid-liquid interface.

Liquid metals have attracted large interest in recent years, not only for fundamental studies[1–3] but also for applications[4–6], such as microfluidics[7,8], electronics[9], reconfigurable devices[10,11], optics[12,13], catalysis[13,14], and materials synthesis[15–18]. Of particular interest is the use of liquid metals in growth processes, for example, in vapor–liquid–solid or electrochemical liquid–liquid–solid deposition[19]. Compared to the use of solid substrates for such processes, liquid metals possess major advantages, which include the high mobility of precursors on the surface, the absence of surface defects such as steps, and the inherent lack of strain, stress, and lattice mismatch that strongly affect the crystal growth process[20]. Moreover, many atomic species can be dissolved in liquid metals, allowing growth from both sides of the liquid interface, and the deposit can be easily separated from a liquid substrate. Recent achievements in this field are the controlled production of

nanostructured materials[6,21], including two-dimensional films[22,23] and crystalline nanowires[15]. Furthermore, the green production of Ge, Si, and GaAs at low temperatures has been demonstrated[15,19,24–29].

Despite these achievements, only little is known about the underlying nucleation and growth processes involved in the deposition at liquid metal interfaces, especially at interfaces to other liquid phases. Most existing studies have been restricted to ex situ characterization of the deposit. This is related to the experimental challenges in obtaining structural data from liquid–liquid interfaces by conventional materials science techniques. As a result, only very few in situ investigations of growth at liquid metal–solution interfaces have been reported so far[30–32]. Among the few structure-sensitive techniques that are capable of accessing liquid–liquid interfaces, X-ray scattering methods, such as X-ray reflectivity (XRR) and grazing

[1]Institute for Experimental and Applied Physics, Kiel University, 24118 Kiel, Germany. [2]Deutsches Elektronen-Synchrotron DESY, Notkestrasse 85, 22607 Hamburg, Germany. [3]Ruprecht-Haensel Laboratory, Kiel University, 24118 Kiel, Germany. [4]Present address: European Synchrotron Radiation Facility, 38000 Grenoble, France. [5]Present address: Mitsubishi Electric Corporation, Advanced Technology R&D Center, 8-1-1 Tsukaguchi-Honmachi, Amagasaki 661-8661, Japan. ✉e-mail: murphy@physik.uni-kiel.de; magnussen@physik.uni-kiel.de

incidence diffraction (GID) measurements, play a prominent role. XRR has been utilized extensively since the 1990s to study the atomic-scale structure of liquid metal–vapor interfaces, for example of Hg[33,34], Ga[35–37], In[38], and liquid alloys[25,39]. These studies showed that liquid metals provide a nearly atomically flat substrate with pronounced atomic layering in the near-surface region. More recent applications of XRR to liquid metal–electrolyte interfaces confirmed these general features of the liquid metal in the presence of aqueous electrolytes and provided fundamental insights into the dependence of surface layering and roughness on potential and temperature[3,40].

In this work, we report a systematic in situ X-ray scattering study of the nucleation and growth of lead halide compounds at liquid Hg–electrolyte interface. The synthesis of lead halide compounds, especially perovskites, has lately received great interest, owing to their potential use in highly efficient solar cells[41–44]. In a previous study, our group reported the nucleation and growth of PbFBr on liquid Hg from PbBr$_2$-containing aqueous NaF solution[31,32,45]. This process was induced by electrochemical de-amalgamation of Pb from the Hg electrode and subsequent precipitation of these ionic compounds[31,32]. At potentials negative of the equilibrium potential $E_{Pb/Pb2+}$, Pb$^{2+}$ is first electrochemically reduced to Pb atoms on the Hg surface, which are then dissolved in the Hg bulk, resulting in the accumulation of Pb in the near-surface region of the liquid metal. Upon increasing the potential into the de-amalgamation regime, the dissolved Pb is rapidly released into the electrolyte[31,32]. This release leads to the supersaturation of Pb$^{2+}$ ions in the electrolyte close to the electrode surface, up to a point where the Pb$^{2+}$ concentration exceeds the solubility product of the Pb halide compounds. As a result, nucleation and growth of crystallites of these compounds commence.

Our in situ X-ray scattering studies revealed that the growth of the ionic compound commences with the rapid formation of a highly uniform precursor adlayer with a thickness of one PbFBr unit cell on the Hg surface, followed by growth of c-axis oriented bulk PbFBr crystals on top of it. This surprising nucleation and growth mechanism was observed over a wide range of growth rates[32] and indicates that, similar to epitaxial growth on solid substrates, the initial interactions at the liquid substrate surface can steer the texture and morphology of the deposit. Because of this analogy to solid-on-solid heteroepitaxy, this highly textured nucleation and growth, which in the literature is often referred to as fiber texture[46], was termed quasi-epitaxial growth. It was not clear, however, whether this phenomenon is specific to PbFBr or represents a more general growth mode at liquid metal interfaces. Especially, the specific influence of the two different halide anion species on this process was not clear.

Here we present in situ X-ray scattering results for a series of deposition systems, in which only one halide anion is present at a time. The influence of the halide species is studied by deposition from aqueous electrolytes of composition NaX + PbX$_2$ with X = Br, Cl, F. We investigate by XRR the presence and structure of precursor adlayers and correlate this with GID data on the formed bulk deposit as well as complementary electrochemical data and optical microscopy observations. Our results confirm the important role of the precursor layer in the nucleation and growth process at liquid metal interfaces and shed light on the influence of the halide species on this process.

## Results

### Electrochemical characterization of the Pb halide systems

Cyclic voltammograms (CVs) of freshly prepared samples in 0.01 M NaX + 0.25 mM PbX$_2$ (X = Br, Cl, F) under steady-state conditions are presented in Fig. 1 (solid lines). Here and in the following, potentials are given with respect to the mercury sulfate electrode (MSE). All studied systems feature typical oxidation/reduction currents due to Pb dissolution/deposition, with an equilibrium potential $E_{Pb/Pb2+} \sim -0.80$ V (as estimated from the midpoint potential between anodic and cathodic peaks in the CV). For potentials $E < E_{Pb/Pb2+}$ the Pb$^{2+}$ ions

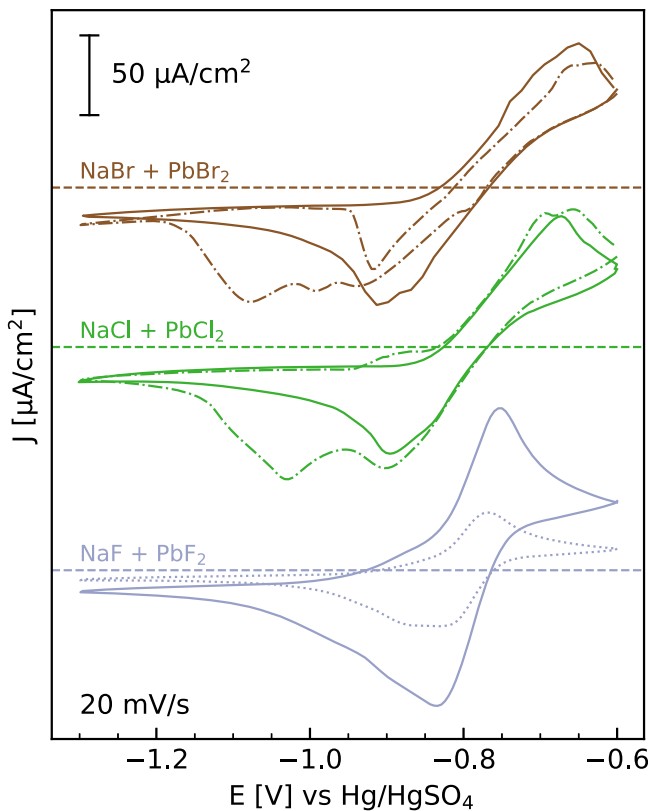

**Fig. 1 | Electrochemical characterization of Pb halide deposition on Hg.** Cyclic voltammograms of Hg in 0.01 M NaBr + 0.25 mM PbBr$_2$ (brown), 0.01 M NaCl + 0.25 mM PbCl$_2$ (green), 0.01 M NaF + 0.25 mM PbF$_2$, (blue solid line) and 0.01 M NaF + 0.12 mM PbF$_2$ (blue dotted line). The CVs (offset for clarity, with current density $j = 0$ indicated by dashed horizontal lines) were measured at 20 mV/s. Solid lines correspond to CVs measured directly after immersion of the electrode into Pb-containing solution, which are stable with time. Dotted dashed lines correspond to the first CV measured after keeping the potential for periods between 60 and 120 min at potentials positive of −0.8 V (120 min at −0.60 V, 60 min at −0.60 V, for Br- and Cl-containing electrolytes, respectively).

amalgamate into Hg as Pb$^0$, whereas for $E > E_{Pb/Pb2+}$ de-amalgamation occurs, where Pb is released into the solution as Pb$^{2+}$. This process and the equilibrium potential do not depend on the halide species. The changes in $E_{Pb/Pb2+}$, the shape of the CV, and the diffusion-limited current density follow the trend, expected for a simple Faradaic reaction.

In CVs in Br- or Cl-containing solution that are measured after holding the potentials positive of $E_{Pb/Pb2+}$ (here, −0.6 V) for 60 to 120 min (dotted-dashed lines), substantial changes are observed. In the presence of Br, the cathodic current in the positive potential sweep exhibits a sharp increase in the range −1.0 to −0.9 V (dependent on sample history), followed by a nearly linear change in the opposite direction. In Cl-containing electrolyte, a shoulder emerges in the positive scan at potentials >−0.95 V. Furthermore, a broad additional cathodic peak is found in the negative sweep at potentials that are ~150 mV more negative than the original amalgamation peak. As shown below, these changes are caused by the formation of Pb halide compounds on the Hg surface in the potential regime of de-amalgamation. Because these ionic compounds dissolve only slowly by a chemical process that is coupled to the local Pb concentration, they partially remain on the surface during the CVs. The additional electrochemical features thus can be attributed to the presence of Pb halide at the interface (see below). In F-containing electrolyte, such effects are not observed (Supplementary Fig. 1). Here, no major changes in the CV are found after longer times at potentials >$E_{Pb/Pb2+}$,

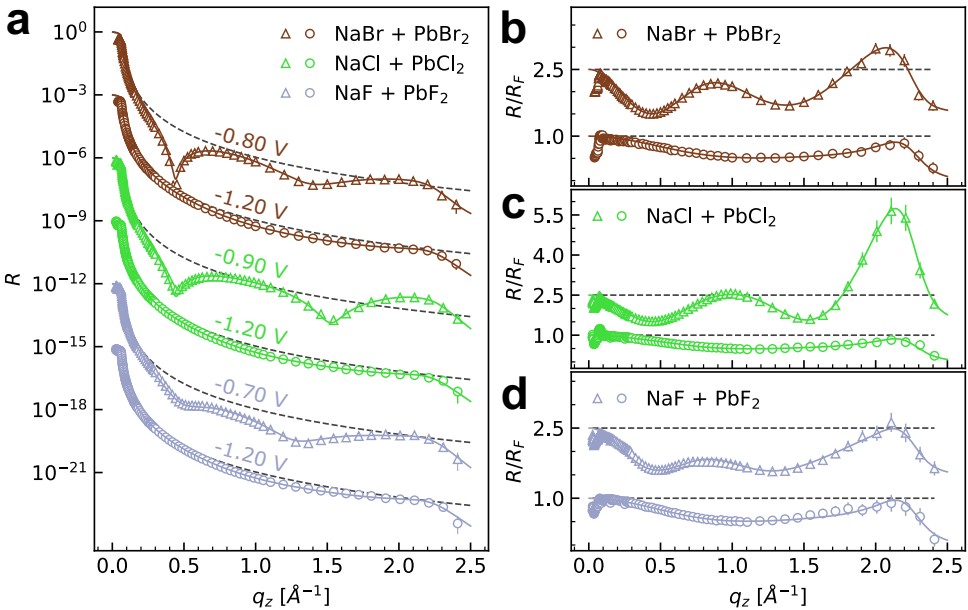

**Fig. 2 | Structural data for the Hg interface structure in Pb halide solutions.**
**a** X-ray reflectivity $R$ and **b**–**d** $R$ normalized by the Fresnel reflectivity $R_F$ (indicated by dashed gray lines) of Hg electrodes in 0.01 M NaBr + 0.25 mM PbBr$_2$ (brown symbols), 0.01 M NaCl + 0.25 mM PbCl$_2$ (green symbols), and 0.01 M NaF + 0.12 mM PbF$_2$ (blue symbols). Triangles indicate curves measured at −1.20 V, open circles curves were measured in the potential range of de-amalgamation, at −0.80 V (brown), −0.90 V (green), and −0.70 V (blue). For clarity, subsequent profiles in **a** are offset by a factor of 1000 and the profiles in **b**–**d** are offset by a factor of 2.5. Solid lines correspond to the best fits with the models discussed in the text. The error bars represent the instrumental errors, resulting from the counting statistics of signal and background.

indicating a fundamental difference from the behavior in Br- and Cl-containing solutions.

## X-ray reflectivity studies of the Hg−electrolyte interface structure

The atomic-scale Hg interface structure was characterized in the different electrolytes by in situ XRR, which depends on the average electron density profile along the surface-normal direction. Figure 2 shows the obtained reflectivity curves of Hg immersed in the three different electrolytes, measured at potentials in the amalgamation (−1.20 V) and de-amalgamation (−0.70 to −0.90 V) regimes. The XRR curves at −1.20 V (triangles) are identical to those measured in Pb-free 0.01 M NaX solution (Supplementary Fig. 2) and are independent of the halide species. They are characteristic of a pristine Hg−electrolyte interface without any adsorbed species from the electrolyte, as has been reported in the previous studies[3,25,31,32,40]. Specifically, the measured reflectivity $R(q_z)$ is close to that of the Fresnel reflectivity $R_F(q_z)$ of a perfectly sharp interface over a wide range of the surface-normal scattering vector $\mathbf{q_z}$, indicating a low interface roughness. With increasing $\mathbf{q_z}$, we observe first an initial decrease of $R(q_z)$ relative to $R_F(q_z)$, followed by an increase towards the so-called pseudo-Bragg peak at ~2.15 Å$^{-1}$. The latter is caused by the well-known stratification of the liquid Hg into atomic layers near its surface[33,47], which decays exponentially into the Hg bulk within 10–20 Å. These data provide clear evidence that the amalgamated Pb atoms diffuse into the bulk liquid metal, leaving the Hg surface unmodified.

Quantitative fits of the XRR interface data at −1.20 V were performed by the distorted crystal model, which is widely used to fit XRR of liquid metal interfaces (details on the fitting process are given in Supplementary Methods 1)[3,31,33,34,36,39,40]. Within the fit errors, the obtained structural parameters (Tables 1, 2, 3) depend neither on the halide species nor on the presence of Pb in the solution. Especially, the Hg layer spacing and the layer decay parameter, which essentially describe properties of the liquid Hg itself, had in all fits identical values of $d = 2.76$ Å and $\sigma_b = 0.48$ Å, respectively. These results indicate that at

this negative potential the halide anion and the amalgamated Pb species do not influence the Hg-electrolyte interface structure, as already shown in our previous studies of the PbFBr system[31].

A very different behavior is found at potentials in the de-amalgamation region (Fig. 2, circles), where the XRR curves deviate strongly from those recorded at similar potentials in the corresponding Pb-free base electrolyte (Supplementary Fig. 2). Here, a pronounced modulation in the range $0 < q_z < 1.8$ Å$^{-1}$ is observed for NaBr + PbBr$_2$, NaCl + PbCl$_2$, and NaF + PbF$_2$. This modulation corresponds to typical "Kiessig fringes"[46], which in XRR experiments indicate the presence of a defined layer on the probed surface. From the position of the first maximum, a characteristic layer thickness in the range of ~7 Å can be estimated. The strong modulation observed in these XRR curves is only possible if the layer contains strongly scattering species, which in this case has to be Pb. All other electrolyte species have scattering length densities that are ~2.3 (Br), ~5 (Cl), and ~10 (F, H$_2$O) times smaller than Pb and thus can be neglected in the first order. These observations provide strong evidence that—similar to the behavior found in our previous studies of the nucleation and growth of the mixed-halide compound PbFBr—a Pb-containing adlayer

## Table 1 | Structural data on the precursor adlayer formed in Br-containing solution

| NaBr + PbBr$_2$ | Parameter | −1.20 V | −0.80 V |
|---|---|---|---|
| Hg | $\sigma_i$ [Å] | 1.01 ± 0.01 | 1.00 ± 0.01 |
| Pb(OH)Br$_{ad}$-I | $\theta_I$ |  | 0.72 ± 0.04 |
|  | $d_{Hg-Br}$ [Å] |  | 2.03 ± 0.10 |
| Pb(OH)Br$_{ad}$-II | $\theta_{II}$ |  | 0.05 ± 0.02 |
|  | $d_{Hg-O}$ [Å] |  | 2.86 ± 0.03 |
| Solution | $\sigma_s$ [Å] | 1.00 ± 0.01 |  |

Results obtained from fits of the XRR data in 0.01 M NaBr + 0.25 mM PbBr$_2$ by the models described in the text. Given are the width of the Hg surface layer $\sigma_i$, the coverages and distance to the Hg surface layer of the precursor adlayer phases Pb(OH)Br$_{ad}$-I and Pb(OH)Br$_{ad}$-II, and the width of the interface to the electrolyte solution $\sigma_s$. The standard errors of the best-fit parameters were calculated by the covariance matrix method.

**Table 2 | Structural data on the precursor adlayer formed in Cl-containing solution**

| NaCl + PbCl$_2$ | Parameter | −1.20 V | −0.90 V |
|---|---|---|---|
| Hg | $\sigma_i$ [Å] | 1.01 ± 0.01 | 0.89 ± 0.01 |
| Pb(OH)Cl$_{ad}$-I | $\theta_I$ | | 0.84 ± 0.02 |
| | $d_{Hg-O}$ [Å] | | 2.63 ± 0.02 |
| Pb(OH)Cl$_{ad}$-II | $\theta_{II}$ | | 0.16 ± 0.02 |
| | $d_{Hg-O}$ [Å] | | 2.63 ± 0.02 |
| Solution | $\sigma_s$ [Å] | | 0.89 ± 0.02 |

Results obtained from fits of the XRR data in 0.01 M NaCl + + 0.25 mM PbCl$_2$ by the models described in the text. Given are the width of the Hg surface layer $\sigma_i$, the coverages and distance to the Hg surface layer of the precursor adlayer phases Pb(OH)Br$_{ad}$-I and Pb(OH)Br$_{ad}$-II, and the width of the interface to the electrolyte solution $\sigma_s$. The standard errors of the best-fit parameters were calculated by the covariance matrix method.

**Table 3 | Structural data on the adlayer formed in F-containing solution**

| NaF + PbF$_2$ | Parameter | −1.20 V | −0.70 V |
|---|---|---|---|
| Hg | $\sigma_i$ [Å] | 1.00 ± 0.01 | 1.03 ± 0.01 |
| Pb Layer 1 | $\rho_1$ | | 0.96 ± 0.06 |
| | $d_{Hg-1}$ [Å] | | 3.10 ± 0.10 |
| | $\sigma_1$ [Å] | | 1.10 ± 0.15 |
| Pb Layer 2 | $\rho_2$ | | 0.47 ± 0.15 |
| | $d_{12}$ [Å] | | 3.09 ± 0.11 |
| | $\sigma_2$ [Å] | | 1.20 ± 0.22 |
| Pb Layer 3 | $\rho_3$ | | 0.13 ± 0.06 |
| | $d_{23}$ [Å] | | 1.35 ± 0.12 |
| | $\sigma_3$ [Å] | | 1.28 ± 0.23 |
| Solution | $\sigma_s$ [Å] | | 1.40 ± 0.28 |

Results obtained from fits of the XRR data in 0.01 M NaF + 0.12 mM PbF$_2$ by the models described in the text. Given are the width of the Hg surface layer $\sigma_i$, the coverages, widths, and distances of the Gaussian layers assigned to Pb, and the width of the interface to the electrolyte solution $\sigma_s$. The standard errors of the best-fit parameters were calculated by the covariance matrix method.

forms on the surface in the de-amalgamation regime in all three studied systems. In addition, in Cl-based electrolytes, the pseudo-Bragg peak's intensity is increased in comparison to the Pb-free electrolyte at this potential (Supplementary Fig. 2), which indicates a reduced interface roughness. This phenomenon was also reported previously in the presence of the PbFBr precursor layer. It can be attributed to a damping of the capillary waves of the interface due to the mechanical rigidity of the precursor layer and indicates that this layer is crystalline.

In the F-containing electrolyte, the oscillations associated with precursor layer formation were less pronounced, indicating a less defined structure. Furthermore, XRR curves with noticeable "Kiessig fringes" could only be observed at a lower PbF$_2$ concentration of 0.12 mM. Even in this electrolyte, however, the XRR curves were not stable in time (on timescales of 1 h) but slowly changed in subsequently recorded XRRs (Supplementary Fig. 3). These changes can be attributed to a slow continuous growth of the deposit, which most likely is limited by the availability of Pb ions. This is supported by the measurements in electrolytes containing 0.25 mM PbF$_2$, where no "Kiessig fringes" but only a bulk deposit could be observed (see below).

**Potentiodynamic changes in the interface structure**
In all systems, the formation of the precursor adlayer is directly linked to the de-amalgamation process. This is illustrated by experiments where the reflected X-ray intensity $I(t)$ was monitored during CVs at a fixed scattering vector $\mathbf{q_z}$ where the XRR curves have a pronounced minimum in the presence of the precursor adlayer (Fig. 3). Upon sweeping the potential towards positive potentials, $I(t)$ starts to drop at ∼−0.98 V in Br-containing, ∼−0.93 V in Cl-containing electrolyte, and ∼−0.80 V (depending on sample history) in F-containing solution, indicating adlayer formation on the Hg surface. This intensity change is

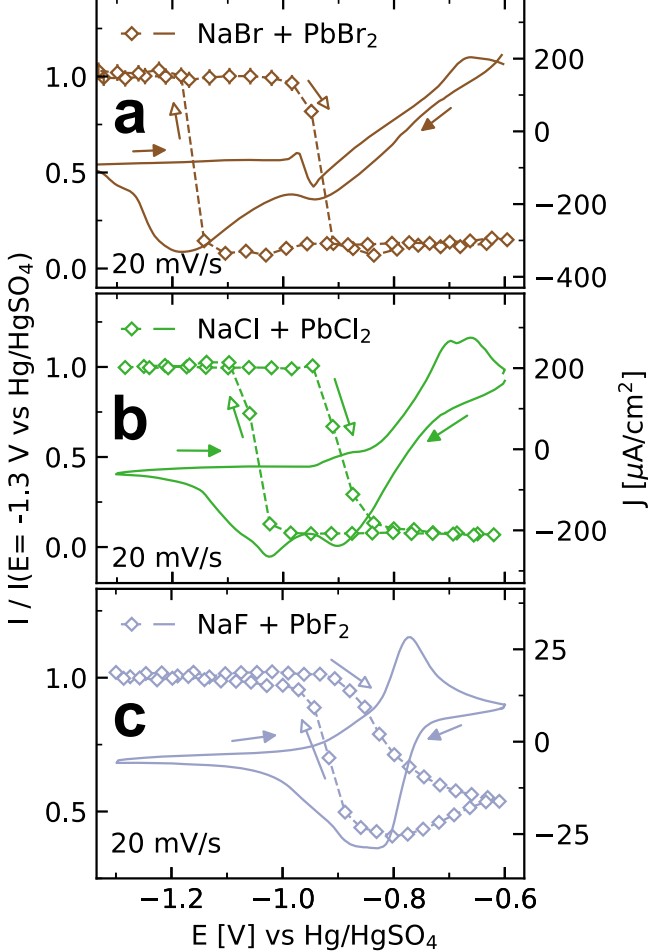

**Fig. 3 | Potentiodynamic changes in the Hg interface structure in Pb halide solution.** X-ray intensity I (open symbols), normalized by the intensity at 1.3 V, at **a** $\mathbf{q_z}$ = 0.50 Å$^{-1}$ and **b**, **c** $\mathbf{q_z}$ = 0.45 Å$^{-1}$, recorded during cyclic voltammograms (solid lines) of Hg at 20 mV/s in **a** 0.01 M NaBr + 0.25 mM PbBr$_2$, **b** 0.01 M NaCl + 0.25 mM PbCl$_2$, and **c** 0.01 M NaF + 0.12 mM PbF$_2$. Arrows indicate the directions of the potential sweeps.

reversible, i.e., the intensity fully recovers when the potential is swept back to a more negative region, albeit with some hysteresis. This reversible behavior indicates that the surface returns to the initial state of a clean Hg surface.

According to the data in Fig. 3, the adlayer formation occurs in different manners and at different rates in the studied systems. In Br- and Cl-containing electrolyte, the drop in X-ray intensity is sharp and occurs within a small potential range of the sweep, corresponding to a time of several seconds. This potential range is in agreement with that of the additional features in the CV that are found after longer residence times in the de-amalgamation region. The good correlation indicates that these features are caused by the presence of the precursor adlayer, which is further supported by the good agreement of the additional cathodic peak in the negative sweep with the change back to the initial X-ray intensity. After each decrease and increase in X-ray intensity, respectively, a defined steady-state intensity is reached, indicating that a defined interface structure has formed and then remains stable.

In F-containing electrolyte, the intensity only starts to decrease positive of −0.90 V and then decreases much more gradually. Here, the intensity of the corresponding XRR curve in Fig. 2 is never reached during the entire CV. Rather, $I(t)$ continues to decrease also in the subsequent negative potential sweep until $E_{Pb/Pb2+}$ is reached, from which $I(t)$ gradually increases again. This behavior indicates that no

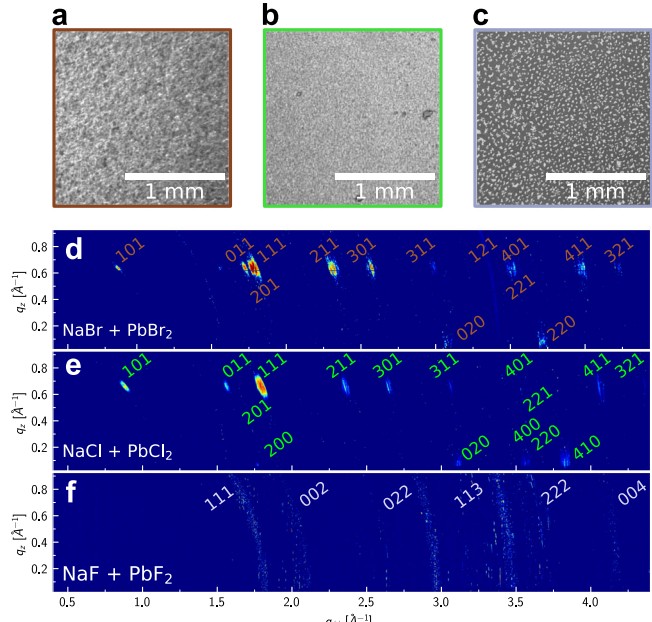

**Fig. 4 | Structure of the bulk Hg halide film. a–c** Optical microscopy images and **d–f** X-ray intensity maps of the deposits on Hg electrodes formed by 1 h deposition at −0.60 V in 0.01 M NaX + 0.25 mM PbX$_2$ with X being **a**, **d** Br, **b**, **e** Cl, and **c**, **f** F, respectively. The GID data were measured at incidence angles of 0.122° and transformed into the reciprocal space coordinates q$_z$ and q$_{//}$. All the experiments were recorded after 1 h deposition at −0.60 V. The miller indexes of the diffraction peaks for the identified compounds **d** Pb(OH)Br, **e** Pb(OH)Cl, and **f** PbF$_2$ are given in the maps.

well-defined precursor adlayer is formed. Instead, the data suggest a slower more continuous layer growth which is in agreement with the XRR observations and the slower growth of the bulk deposit in this system (see below).

The kinetics of the structural transitions at the Hg-electrolyte during such sweeps are complicated and depend on the precise conditions of the potential sweep and the sample history. The pronounced intensity changes in Br- and Cl-containing solution (Fig. 3a, b) are only observed in potential sweeps where the sample was previously kept in the de-amalgamation regime. For CVs, starting on the pristine Hg electrode, corresponding intensity changes were found at similar potentials but of much smaller magnitude (Supplementary Fig. 4). Furthermore, in the transition region, $I(t)$ occasionally exhibited strong fluctuations (Supplementary Fig. 5). Similar observations were made in PbFBr nucleation and growth and could be explained by the lateral movement of deposit islands floating on the liquid Hg surface and electrocapillary effects, resulting in long-range mechanical deformation of the liquid electrode[32].

### Growth of bulk deposits at the liquid metal interface
Before we investigate the atomic-scale arrangement within the adlayer in more detail, we first discuss observations on thicker deposits, as they provide valuable clues on the composition and structure of the adlayer phases. Such deposits were formed after holding the potentials positive of $E_{Pb/Pb2+}$ for 60–120 min and could be optically detected. Characteristic microscope images of the deposited film are shown in Fig. 4a–c. Clear differences in the deposit morphology formed in the three systems under identical conditions are found. In Br-containing electrolyte (Fig. 4a), the deposit appears rough but uniform; in Cl-containing solution (Fig. 4b), a smoother film is formed. In contrast to the two former cases, the deposit grown in F-containing electrolyte (Fig. 4c) consists of isolated micrometer-sized crystals. From the image, an average crystal size of ~35 μm was estimated. The individual

crystals did not change their positions in subsequent microscope images, indicating that they were not mobile on the liquid surface but fixed in position relative to each other. This is attributed to the presence of a surrounding much thinner layer that cannot be detected by optical microscopy—most likely the initial adlayer. We note that no visible bulk deposit (based on optical microscopy and GID data) could be observed in electrolytes containing only 0.12 mM PbF$_2$.

XRR studies of these bulk deposits were not possible due to the high interface roughness. The deposits were therefore characterized by in situ GID measurements, whose results are displayed in Fig. 4d–f. For Br- and Cl-containing electrolytes, discrete Bragg peaks are observed, indicating that the films consist of crystals with a well-defined orientation along the surface-normal direction (see Supplementary Fig. 6)[32]. In contrast, only powder rings are detected for deposits grown in F-containing electrolyte, indicating a random orientation of the formed crystals. By integration of the detector intensities at identical $\mathbf{q} = \sqrt{\mathbf{q_z^2 + q^2}}$, with $\mathbf{q}_{||}$ being the lateral wave transfer vector, powder spectra $I(\mathbf{q})$ were generated (Supplementary Figs. 7–9), which were then used to identify the composition and structure of the crystallites. For this, the diffraction peak patterns were compared to the crystal structures of Pb halides and other Pb-containing compounds (Supplementary Figs. 7–9).

Based on this analysis, the crystal structures of the deposits in Br- and Cl-containing electrolytes could be unambiguously identified. For the case of Br, an excellent match with orthorhombic Pb(OH)Br (pnma, with $a = 7.385$ Å, $b = 4.085$ Å, $c = 10.012$ Å, ICSD 404573) was found (Fig. 5a). Pb(OH)Br formation in this solution was already reported in similar studies[48]. For Cl-containing solution, orthorhombic Pb(OH)Cl (pnma, with $a = 7.111$ Å, $b = 4.020$ Å, $c = 9.699$ Å, ICSD 28035) provides a good match to the GID data[49,50]. Orthorhombic Pb(OH)Br and Pb(OH)Cl exhibit the same crystal structure (laurionite), apart from small differences in the bond lengths due to the different halide species. The rather slow growth of these deposits may be related to kinetic limitations caused by the low OH⁻ concentration in the employed neutral electrolytes. The structure formed in F-containing electrolyte is less clear. Cubic PbF$_2$ (fm3m, with $a = b = c = 6.002$ Å, ICSD 201114) shows a possible match but not all observed peaks could be attributed to this compound. Probably a hybrid Pb(OH)F and/or Pb$_x$O$_y$H$_z$ structure is present given the lower Gibbs energy for oxides and hydroxides[51]. However, in contrast to the case of Pb(OH)Cl and Pb(OH)Br, there is no Pb(OH)F structure reported in the literature up to date.

With the identified crystal structure in Br- and Cl-containing electrolytes, also the orientation of the crystals along the surface normal can be determined from the intensity maps in Fig. 4d–f. Here, all Bragg peaks were found at $\mathbf{q_z}$ values of 0 (partly cut off by the Hg substrate) and at ~0.64 Å⁻¹. The latter value is in excellent agreement with that of the (001) peaks in Pb(OH)Br ($\mathbf{q_z} = 0.63$ Å⁻¹) and Pb(OH)Cl ($\mathbf{q_z} = 0.65$ Å⁻¹), indicating a film texture where the $c$-axis is oriented along the surface normal. All experimentally observed peaks can be assigned to the (hk0) and (hk1) peaks of crystals in this orientation (see labels in Fig. 5d, e).

### Structure of the adlayer formed in the de-amalgamation regime
The results obtained on the structure and orientation of the bulk deposits provide a valuable starting point for the determination of the precise structure of the precursor adlayer from the XRR data. Specifically, we assume in the models employed to fit the XRR data that the precursor adlayer consists of the same material and has the quasi-epitaxial orientation as the bulk deposit, as was found in our previous studies of PbFBr growth[31,32]. In the following, the results obtained for the three systems are described in the order Br-, Cl-, and F-containing electrolyte. In all fits, the Hg layer spacing $d$ and the layer decay parameter $\sigma_b$ were kept constant to reduce the number of free fit parameters.

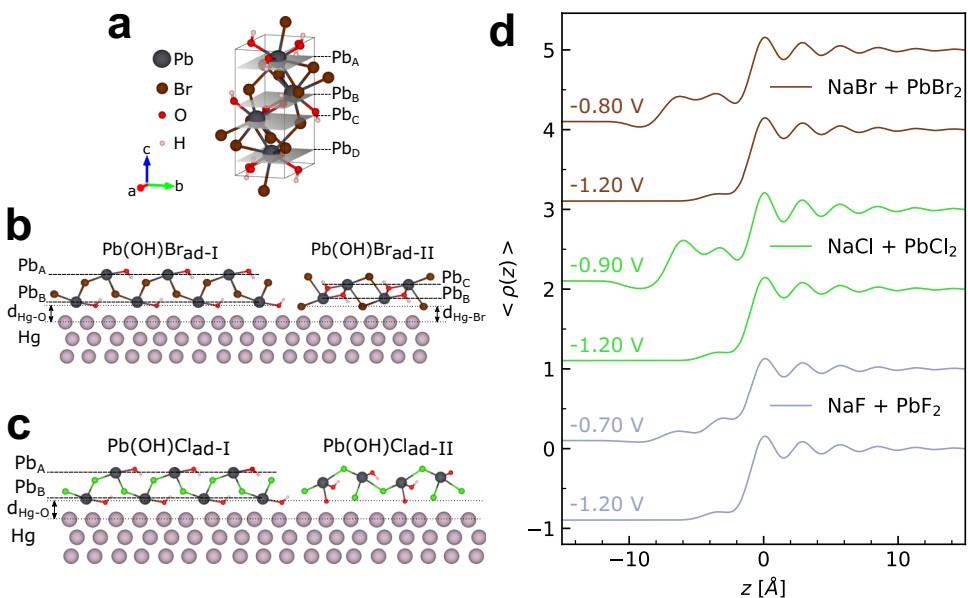

**Fig. 5 | Bulk and interface structure of the PbBr₂ and PbCl₂ systems. a** Laurionite crystal structure, illustrated for the case of Pb(OH)Br (generated by VESTA[72]). **b** Profiles of the electron density $\langle \rho(z) \rangle$ obtained from fits of the XRR measurements in Fig. 2, corresponding to the interface structure in the potential regime of Pb de-amalgamation in Br (brown), Cl (green), and F (blue) containing electrolyte. **c, d** Structural models used to describe the Hg-electrolyte interface in the presence of Pb-containing adlayers for the case of **c** Br and **d** Cl anions.

For Br-containing electrolyte, we assume that the precursor adlayer consists of c-axis oriented Pb(OH)Br (Supplementary Methods 2 and Supplementary Fig. 10). However, the modulation period $\Delta \mathbf{q_z} \sim 0.95\,\text{Å}^{-1}$ in the XRR curves corresponds to a thickness of roughly $d = 2\pi/\Delta \mathbf{q_z} \sim 6.6\,\text{Å}$, which is clearly smaller than a complete Pb(OH)Br unit cell ($c = 10.012\,\text{Å}$) and indicates that the adlayer is only composed of a fraction of the unit cell. Preliminary fits show clearly that good fits of the XRR data require two pronounced peaks in the electron density profile of the adlayer (Fig. 5b), which we attribute to 2 atomic planes containing Pb atoms. For a more precise structure determination, various models and structures were tested but only one was able to provide fits that are in excellent agreement with the experimental data (for fits by different models see Supplementary Methods 2). In this successful model, the majority of the adlayer consisted of a section of the Pb(OH)Br unit cell where the Pb planes denoted as A and B (see Fig. 5c and Supplementary Fig. 10) correspond to the electron density peaks in the adlayer (denoted as Pb(OH)Br$_{ad}$-I in Fig. 5c). Fits with this adlayer phase alone could not fully describe the data (Supplementary Fig. 11) but a second minority adlayer phase with a relative surface coverage of only 7% as compared to the majority phase was required (denoted as Pb(OH)Br$_{ad}$-II). This second phase had a different termination at the Hg surface, resulting in a smaller vertical distance between the two Pb layers.

The structural parameters obtained for the best fit by this two-phase model are given in Table 1. Here, terminating anion layers of OH and Br were assumed in Pb(OH)Br$_{ad}$-I and Pb(OH)Br$_{ad}$-II, respectively (see Fig. 5c). For Pb(OH)Br$_{ad}$-I, the distance to the Hg substrate was found to be $d_{Hg\text{-}O} = 2.86 \pm 0.03\,\text{Å}$, which is comparable to typical Hg-O distances of $2.2$–$2.9\,\text{Å}$[52,53]. The distance for the halide-terminated Pb(OH)Br$_{ad}$-II phase was found to be $d_{Hg\text{-}Br} = 2.03 \pm 0.10\,\text{Å}$. This value is slightly smaller than values reported in the literature ($2.4$–$2.5\,\text{Å}$)[54,55]. It is noted that the presence of these terminating OH or Br anions cannot be directly verified, because their contribution to the total electron density profile is small. Considering only the spacing between the first Pb and the Hg surface layer, which is a robust parameter in the fits, rather similar values are found for the two adlayer phases: $3.30\,\text{Å}$ in Pb(OH)Br$_{ad}$-I and $3.00\,\text{Å}$ in Pb(OH)Br$_{ad}$-II. These distances are sufficiently short to allow also for direct bonding

of the Pb atoms to the Hg surface. The total surface coverage of the precursor layer (Pb(OH)Br$_{ad}$-I and Pb(OH)Br$_{ad}$-II) is 77% of the whole Hg electrode for the data shown in Fig. 2 but varied by ±10% in different experiments. Such an incomplete coverage of the Hg surface was also observed for the case of PbFBr precursor adlayers[31,32]. In the remaining surface area, the Hg could either be free of the deposit or covered by three-dimensional Pb(OH)Br bulk deposits, which would not contribute to the features in the XRR. Considering that the measurement of an XRR curve requires 60 min, the latter appears more likely.

For the adlayer formed in Cl-containing electrolyte, analogous results were obtained (Table 2). Also here, a good fit was only possible by a mixture of two different surface phases (see Supplementary Methods 2). The phase with the larger coverage (denoted Pb(OH)Cl$_{ad}$-I) corresponds to the majority phase found in the Pb(OH)Br adlayer, i.e., c-axis oriented Pb(OH)Cl that is terminated by OH at the Hg surface (see Fig. 5d and Supplementary Fig. 12). In contrast to the case of Pb(OH)Br, the second phase cannot be described by a similar c-axis oriented Pb(OH)Cl phase with different surface termination. In particular, all successful fits of the corresponding XRR require that the peak in the electron density profile facing the electrolyte is more pronounced than the peak facing the Hg surface (see Fig. 5b and Supplementary Fig. 13). This can be accomplished by structures where the outer peak is composed of two closely spaced Pb layers. A phase that would fulfill this requirement is a Pb(OH)Cl adlayer that is oriented with the a-axis along the surface normal (Fig. 5d, Pb(OH)Cl$_{ad}$-II). Excellent fits with this model are possible, with the distance between the terminating OH layer and the Hg surface layer being identical to that in Pb(OH)Cl$_{ad}$-I. However, also models where this Pb(OH)Cl$_{ad}$-II minority phase contains an additional Pb layer at the Hg surface (with $d_{Pb\text{-}Hg} = 2.83\,\text{Å}$) or have different Hg-O distances describe the data well. It is further noted that a good fit can also be obtained by models that assume other crystalline materials as the minority phase, especially c-axis oriented PbO(H₂O). However, the GID data exclude the presence of a similarly oriented PbO(H₂O) bulk phase. Furthermore, the formation of a PbO(H₂O) adlayer would also be expected in the Br- and F-containing electrolytes, which was not observed. Thus, this alternative explanation does not seem likely.

Description of the adlayer formed in 0.1 M NaF + 0.12 mM PbF$_2$ was even more difficult than in the Cl- and Br-system. Since the crystal structure of the formed bulk deposit is unclear and the deposit is powder-like, it was not possible to model the adlayer by assuming the same composition and orientation as within the three-dimensional crystals. We therefore employed a more general approach describing the adlayer by simple Gaussian peaks in the electron density profile without any assumption of the corresponding chemical species. As discussed above, the main contribution to these peaks will come from the strongly scattering Pb atoms, whereas the contributions of O and F will be negligible. We thus denote these Gaussian peaks in the following as Pb layers. Reasonable fits of the experimental XRR curves could only be obtained by a series of three Pb layers that decayed in intensity (by a factor >7 from the first to the third Pb layer) and increased in width from the Hg surface towards the electrolyte (see Fig. 5b, Table 3, and Supplementary Fig. 14).

Deconvolution of the Pb layer by the width of the Hg surface layer $\sigma_i$, which is dominated by the capillary wave width, results in effective widths of 0.34 Å for the first, 0.62 Å for the second, and 0.76 Å for the third Pb layer. These observations indicate that in contrast to the Pb(OH)Br and Pb(OH)Cl adlayers, the ordering along the surface normal direction rapidly decays. The distance of the first Pb layer to the Hg surface (3.10 Å) is comparable to that in the two other systems. The distance between Pb layer 1 and 2 (3.09 Å) differs by only 3% from that of the Pb planes in a PbF$_2$ cubic structure, but the much shorter distance between Pb layers 2 and 3 cannot be explained by a PbF$_2$ phase. However, the two distances are similar to those found between the Pb layers in Pb(OH)Br and Pb(OH)Cl. The observed adlayer could hence correspond to a mixture of PbF$_2$ and of Pb(OH)F with laurionite crystal structure, although the latter phase has hitherto not been observed in the bulk.

As mentioned above, only the vertical position of the first Pb layer is reasonably well localized, whereas the following layers already have a rather large width. Thus, a coexistence of crystallites with different orientations is possible, which would be in agreement with the powder-like bulk deposit structure in F-containing electrolyte. In total, the XRR data in electrolytes containing exclusively F anions suggest that instead of a well-defined precursor adlayer only thin crystallites of a three-dimensional deposit are formed in a solution containing 0.12 mM PbF$_2$. This is supported by experiments showing that subsequently recorded XRR curves in this solution are not stable but continuously change with time (Supplementary Fig. 3). This indicates slow, kinetically limited growth of the deposit, which is in accordance with the absence of bulk crystallite formation in the optical microscopy and GID studies. At higher PbF$_2$ concentration, no distinct adlayer formation but only bulk deposition was found, which further supports this scenario.

## Discussion

The structural data obtained from the in situ X-ray scattering studies reveal fundamental differences in the phase formation processes found in the presence of Br and Cl anions as compared to those in the presence of F anions alone[31,32].

In principle, this growth occurs by a purely chemical mechanism in solution and the main role of the electrochemical process is to provide a sufficient local Pb$^{2+}$ ion concentration. These ions are continuously released in the de-amalgamation potential regime, leading to a stabilization of the Pb halide bulk deposits. In the absence of specific interactions with the Hg-electrolyte interface, one would therefore expect a randomly oriented, powder-like deposit. This type of behavior seems indeed to be the case in NaF + PbF$_2$ solution, where our data indicate direct nucleation and growth of poorly ordered, three-dimensional crystallites. In the presence of Cl or Br anions in solution, however, Pb halide growth proceeds by a different route, as shown by our observations in NaCl + PbCl$_2$ and NaBr + PbBr$_2$ solution

as well as our earlier results in NaF + PbBr$_2$ electrolyte[31,32]. Specifically, we find in all these systems a well-defined texture of the bulk deposit, where the crystals are oriented with the c-axis parallel to the surface normal direction. This quasi-epitaxial growth indicates a steering role of the Hg substrate in the crystal growth.

The XRR measurements demonstrate that the occurrence of oriented crystal growth is strongly linked to the formation of well-defined precursor adlayers in the initial stages of the deposition process. For Pb(OH)Br and PbFBr formation, the precursors are c-axis oriented crystalline films with a thickness of two Pb atomic layers, i.e., two formula units of the Pb halide compound. The subsequent quasi-epitaxial growth of c-axis oriented Pb(OH)Br and PbFBr, respectively, merely constitutes a continuation of these sub-nanometer-thick crystals and thus is easily explained. For Pb(OH)Cl growth, the same reasoning holds true for the majority adlayer phase Pb(OH)Cl$_{ad}$-I, which is structurally analog to the Pb(OH)Br$_{ad}$-I phase. The further vertical growth of the minority phase Pb(OH)Cl$_{ad}$-II seems to be kinetically hindered, as bulk crystallites with the same orientation as in Pb(OH)Cl$_{ad}$-II were not observed in the GID measurements. The precise reasons for that are not clear at present and require further studies. Based on the microscopy investigations, which show that the entire Hg surface is covered by the Pb(OH)Cl film after longer deposition times, the surface areas covered by the Pb(OH)Cl$_{ad}$-II phase are either converted into Pb(OH)Cl$_{ad}$-I or overgrown by lateral growth of the oriented Pd(OH)Cl bulk crystals. That certain nuclei with preferred crystal orientations can grow on cost of less favorable oriented nuclei is a well-known phenomenon in thin film deposition and encountered for example in columnar growth[56].

Having established that the existence of well-oriented precursor adlayers is the reason for quasi-epitaxial growth, we now address the origin of precursor adlayer formation. According to our data, these adlayers form in the presence of Cl and Br anions but not in electrolytes containing only F (and OH) anions. Chloride and bromide are known as electronically polarizable (soft) anion species that strongly chemisorb on metal surfaces. Specifically, the specific adsorption, i.e., chemical binding, of Cl and Br on Hg electrodes is known since the early days of interfacial electrochemistry[57–59]. In contrast, fluoride is a hard anion with a strongly bound hydration shell and known for its non-specific adsorption on Hg. Consequently, F anions only reside as hydrated, electrostatically bound species in the outer Helmholtz plane, whereas Cl and Br form chemisorbed adlayers on Hg, which can reach considerable coverages at potentials positive of the potential of zero charge ($E_{pzc}$ = −0.85 V[57], −0.89 V[60], and −0.91 V[57] vs. Hg/HgSO$_4$ in the F-, Cl-, and Br-containing electrolyte, respectively). In our previous studies[31,32], we proposed that these chemisorbed anions stabilize coadsorbed Pb$^{2+}$ counter ions on the surface. The latter carries twice the charge as the halide and thus attracts further ions until charge neutrality is reached after a total of five atomic layers, with the last one being a terminating anion layer.

For the systems studied in the present work, the scenario is in principle similar but the detailed explanation of the formed adlayer structure is more difficult. This is mainly caused by the more complex crystal structure of Pb(OH)Br and Pb(OH)Cl, which result from the presence of OH and cannot be understood from simple isotropic ion–ion interactions alone. As an example, we briefly consider the majority adlayer phases, which will be most important phases for the subsequent quasi-epitaxial growth. Here, the Pb atoms in the two cation layers are interlinked by a buckled layer, composed of Br anions with an in-plane square arrangement (Fig. 5c and Supplementary Fig. 10). This structure results from the space requirements of the Br-bound OH groups, which are located slightly above and below the Pb planes, respectively. The inner Pb layer appears to be directly adsorbed on the Hg surface layer rather than bound via an intermediate halide adlayer. In contrast, the minority phase Pb(OH)Br$_{ad}$-II involves adsorbed Br at the liquid Hg electrode.

Obviously, the stabilization of defined precursor adlayers by coadsorbed cations is possible in a variety of atomic arrangements on the liquid metal surface. This is in agreement with the well-known formation of mixed chemisorbed layers of anions and cations on solid metal electrodes (see ref. 61 for an overview). In previous work, it has been shown that these species can mutually stabilize themselves on the electrode surface. Particularly insightful are in situ X-ray surface scattering studies of Wang and co-workers that reported the presence of mixed adlayers of halides with Tl or Pb on Au(111) electrodes in acidic electrolyte[62–64]. These systematic studies reported a complex potential-dependent sequence of differently ordered adlayer structures, in which the stoichiometry changed stepwise from a purely metallic adlayer to a pure halide layer. In an intermediate potential range, various mixed phases were found, including phases with a square 1:1 arrangement of halide and metal. The halide was either coadsorbed with the metal species on the Au surface or resided on top of the adsorbed metal layer. In both cases, the ad-metal is stabilized on the electrode surface at much more positive potentials as in halide-free electrolyte. For Pb in Br-containing solution, the transition between the different mixed adlayer phases was slow, resulting in long-term coexistence of those phases[64]. All these observations closely resemble our findings for the precursor adlayers on liquid Hg electrodes. The main differences are the presence of a second anion species in the adlayer (F or OH) and its larger vertical extension, i.e., the presence of 2 Pb and several anion layers. The former difference can be related to the coexistence of two anions in the electrolyte and an apparently lower Gibbs energy of formation of these compounds as compared to more simple salts, such as $PbBr_2$ or $PbCl_2$[31]. The larger vertical extension was attributed in ref. 31 to electrostatic effects, resulting from the different charges of the $Pb^{2+}$ and the monovalent anions and the need to achieve charge neutrality in the adlayer. Similar reasoning can be applied to the precursor adlayer phases observed in this study.

Within the scenario described above, the tendency towards precursor adlayer formation should increase with increasing Gibbs energy of adsorption of the halide on the Hg electrode, i.e., in the order F < Cl < Br. This is indeed found in potential-dependent XRR studies, which show that both formation and dissolution of the adlayer start to shift in the same order towards more negative potentials. For example, the onset of dissolution in the negative potential scans of the experiments in Fig. 3 starts in F-containing solution at −0.80 V, i.e., directly at $E_{Pb/Pb2+}$, in Cl-containing solution at −1.0 V, and in Br-containing solution at −1.1 V. This supports our assumption that chloride and bromide stabilize Pb-containing precursor adlayers on the Hg surface in a wide potential range, from the Pb amalgamation up to the de-amalgamation regime, where Pb dissolution into the bulk Hg or $Pb^{2+}$ desorption would be expected, respectively. In contrast, fluoride anions do not lead to any stabilization and therefore only bulk precipitates without a defined orientation towards the interface are formed. The kinetics of adlayer formation and dissolution exhibits this trend too, resulting in faster and more facile formation with increasing chemisorption strength of the halide anion.

Our in situ X-ray scattering studies show that, in a deposition at liquid metal−electrolyte interfaces, chemisorbing ions, such as chloride and bromide, can serve as surfactants that promote the growth of strongly textured deposits. This behavior can be linked to the formation of well-defined sub-nanometer thick precursor adlayers of the deposited compounds that serve as nuclei for the subsequent quasi-epitaxial growth of oriented crystals of the bulk phase. The mechanistic scenario proposed for this process should be valid for many materials, provided they contain species that chemisorb at the interface but are poorly solvable in the liquid metal. The stability and ease of formation of the precursor adlayers scale with the strength of the chemisorption. Even stronger tendencies for precursor adlayer formation and quasi-epitaxial growth are therefore expected in the presence of iodide or sulfide anions. This opens up interesting possibilities

for the preparation of technologically interesting compound semiconductors. Already the Pb(OH)Br phase is interesting in that respect, as it was shown to be chemically convertible into a perovskite material for resistive memory devices[65]; for PbI perovskites an even larger range of potential applications exists. Furthermore, the same principles should be applicable to the growth of compounds with other cations. For example, Cs and Cu are well known for forming salt-like adlayers with halides on solid electrodes[61] and thus are likely candidates for similar quasi-epitaxial growth. In addition, these processes might be transferable to other liquid metal substrates[19], such as InHg alloys[25], or less toxic Ga[28,66,67] which likewise provide atomically smooth substrates and have been already employed in the electrochemical growth of a range of materials. Due to the liquid electrode, upscaling the synthesis for industrial growth is a realistic possibility[68].

The growth-steering effects at interfaces to liquid metals and the resulting deposit texture are stronger than those found at other liquid-liquid interfaces, such as interfaces between aqueous and non-aqueous solutions[69]. One important reason for this is the strong chemical interactions of many species with metal surfaces. These chemical interactions lead to strongly bound adsorbate layers with a high surface coverage, which promote the formation of oriented precursor adlayers. In addition, the high surface tension of liquid metals results in atomically smooth interfaces, providing ideal substrates. Because of these properties, growth at liquid metal interfaces is an interesting route for the formation of crystalline nanomaterials and thin films. The mechanistic insights obtained in this work may help in developing these deposition processes and tuning them, e.g., toward higher crystallinity and optimized morphology. Specifically, precursor-mediated growth may also allow to further improve emerging methods employing liquid metals, such as electrochemical liquid-liquid-solid deposition.

## Methods

### X-ray scattering measurements

X-ray reflectivity and GID studies were performed using the LISA liquid interface diffractometer[70] at beamline P08 of PETRA III (DESY)[71]. X-ray energy of 25 keV was chosen to have acceptable counts despite the strong X-ray absorption by the electrolyte. All data were collected with a two-dimensional X-ray detector with 55 μm pixel size (X-Spectrum Lambda 750k GaAs). The acceptance of the detector to integrate the intensity for XRR was 0.35° vertically and 0.06° horizontally, the background subtraction was calculated by offsetting the region of interest (ROI) horizontally with the same acceptance by 0.06°. For the GID studies of the bulk deposit structure, the incident angle was fixed at 0.122°, corresponding to 85% of the critical angle for the system. An in-plane scan of the detector with 50 images was recorded (with 30 s counting time, 0.40° intervals). From each image, the background, obtained by an identical scan performed before the deposition, was subtracted, and the resulting 50 images were then binned with a $q_{//}$ resolution between $5 \times 10^{-3}$ and $5 \times 10^{-4}$ Å$^{-1}$.

### Sample environment and samples

The experiments were performed in a dedicated electrochemical sample environment, which is described in detail in Supplementary Methods 3. It consisted of a sealed PEEK cell, with an upper quartz window that provides optical access for the microscope camera (Basler ACE acA1600−20gm with OPTEM Fusion motorized Zoomsystem). The cell contains a liquid mercury pool (Chempur, 99.999+%) of 23.7 cm$^2$ surface area with a Pt wire contact and approximately 55 mL of the electrolyte. A Hg/Hg$_2$SO$_4$ (Schott) reference electrode, a Pt counter electrode, and an Ametek Solartron Analytical potentiostat (Modulab 2101A) were used for potential control. All potentials mentioned in this publication are given relative to Hg/Hg$_2$SO$_4$. The sample cell parts that were in contact with the sample were previously soaked for at least 6 h in caroic acid, a 3:1 solution of sulfuric acid (Carl Roth,

ROTIPURAN® 96%, p.a., ISO) and hydrogen peroxide (Carl Roth, ROTIPURAN® 30%, p.a., ISO), and then sonicated and rinsed with Milli-Q water prior to sample preparation. To clean the PEEK components a 20–30% caroic solution was used to avoid dissolution of the cell material. The electrolyte solutions were prepared from NaF (Alfa Aesar Puratronic®, 99.995%), NaCl (ChemPur, 99.999%), NaBr (Merck Suprapur® 99.995%), $PbF_2$ (abcr, Puratronic®, 99.997%), $PbCl_2$ (Chem-Pur, 99.999%), $PbBr_2$ (Alfa Aesar Puratronic®, 99.998%), and Milli-Q water. They were purged with $N_2$ gas for at least 30 min before injecting them into the cell to remove $O_2$ dissolved in the electrolyte. After initial sample characterization in the pure NaX base electrolyte, the electrolyte was exchanged by flowing 500 mL of NaX + $PbX_2$ solution through the cell, i.e., more than 9 times the cell volume. All experiments were performed at room temperature (22.5 °C).

## Data availability
The data used to generate the figures shown in this study have been deposited in the Zenodo database https://doi.org/10.5281/zenodo.6940516.

## Code availability
The custom software for the analysis of the XRR data is available in the repository https://doi.org/10.5281/zenodo.6940544.

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

## Acknowledgements

The authors acknowledge financial support by the DFG through project MA 1618/18 (O.M.M.) and funding for LISA by the BMBF through project 05K16FK1/05K19FK2 (O.M.M., B.M.M.). We acknowledge DESY (Hamburg, Germany), a member of the Helmholtz Association HGF, for the provision of experimental facilities. Parts of this research were carried out at PETRA III and we would like to thank Florian Bertram and Rene Kirchhof for assistance in using P08 and Milena Lippmann for her assistance in using the PETRA III chemistry laboratory. Beamtime was allocated for proposals I-20200512, I-20180496, and II-20170014. Furthermore, we would like to thank Wulf Depmeier for his feedback on the laurionite crystal structure.

## Author contributions

A.S., R.P.G., H.F., S.C.H., J.E.W., and P.J., performed the experiments. C.S. assisted during beamtime as a beamline scientist. A.S. performed the data analysis. A.S., B.M.M., and O.M.M. interpreted the data and wrote the manuscript.

## Funding

## Competing interests

The authors declare no competing interests.
