## [Peer Review File · Nature Communications]

Reviewer comments, first round

Reviewer #1 (Remarks to the Author):

Review of "Role of chemisorbing species in growth at liquid metal-electrolyte interfaces revealed by in-situ X-ray scattering" by Sartori et al.

This is a nice paper by a very capable group known for their studies of immersed interfaces. The paper is pretty much good to go although the authors may wish to consider the following comments before moving to final publication.

The paper would benefit from moving, or adding, some material on supersaturation that is present in the first paragraph of the discussion section to the last paragraph of the introduction. This will give the reader better context and appreciation for the experiment that is to be detailed thereafter.

The condition associated with generating the voltammetry in Figure 1 needs a clearer description. For the solid lines where were the voltammograms initiated or are we looking at a "steady-state" result. Likewise for the dotted lines after aging at potentials positive of -0.8 V? How far positive, i.e. if too positive presumably all the Pb would be removed, can the authors please be more precise. The overlayer derives from supersaturationpresumably the system should relax and eventual dissolve or is it the slow or steady supply of Pb from the mercury pool that enables the discrete and stable ultrathin overlayer observed in Fig 3 and b?

I was somewhat surprised by the hydroxide component in the adlayer and crystals. What is the solution pH and if the pH is 8 or less shouldn't we expect the kinetics of crystallization to be limited by the available OH-?

Rather than "quasi-epitaxial" the authors might wish to refer to "highly textured nucleation and growth".

How important is the 0.01 mol/L Na⁺ addition to the present work. Did the authors consider its possible role in the adlayer structures.

For Hg novices the authors may wish to make references to the pzc of Hg and Pb at some point in the paper. The discussion section otherwise is excellent.

Reviewer #2 (Remarks to the Author):

This is a very interesting and detailed work on using various techniques, including in situ X-ray scattering studies, to demonstrate that deposition at liquid metal – electrolyte interfaces the chemisorbing ions, such as chloride and bromide, can serve as surfactants that promoting the growth of strongly textured precipitates. I do not have any technical comments on the work, but would ask the authors to comment on how this method can be generalized to other materials (in addition to the toxic ones discussed here) and what concrete role for an upscaled synthesis this approach can have.

Response to reviewers

Reviewer 1

Review of "Role of chemisorbing species in growth at liquid metal-electrolyte interfaces revealed by in-situ X-ray scattering" by Sartori et al.

This is a nice paper by a very capable group known for their studies of immersed interfaces. The paper is pretty much good to go although the authors may wish to consider the following comments before moving to final publication.

The paper would benefit from moving, or adding, some material on supersaturation that is present in the first paragraph of the discussion section to the last paragraph of the introduction. This will give the reader better context and appreciation for the experiment that is to be detailed thereafter.

We agree with the reviewer and now discuss this issue at the end of the introduction, rather than in the discussion.

Corresponding changes in manuscript (p. 3):

This process was induced by electrochemical de-amalgamation of Pb from the Hg electrode and subsequent precipitation of these ionic compounds^{32,33}. At potentials negative of the equilibrium potential $E_{Pb/Pb^{2+}}$, Pb^{2+} is first electrochemically reduced to Pb atoms on the Hg surface, which are then dissolved in the Hg bulk, resulting in accumulation of Pb in the near-surface region of the liquid metal. Upon increasing the potential into the de-amalgamation regime, the dissolved Pb is rapidly released into the electrolyte^{32,33}. This release leads to supersaturation of Pb^{2+} ions in the electrolyte close to the electrode surface, up to a point where the Pb^{2+} concentration exceeds the solubility product of the Pb halide compounds. As a result, nucleation and growth of crystallites of these compounds commence.

The condition associated with generating the voltammetry in Figure 1 needs a clearer description. For the solid lines where were the voltammograms initiated or are we looking at a "steady-state" result. Likewise for the dotted lines after aging at potentials positive of -0.8 V? How far positive, i.e. if too positive presumably all the Pb would be removed, can the authors please be more precise.

We thank the reviewer for this helpful comment and now provide detailed information about the conditions used for recording the CVs in the figure caption of Fig. 1.

Corresponding changes in manuscript (caption of Fig. 1):

The CVs (offset for clarity with $j = 0$ indicated by dashed horizontal lines) were measured at 20 mV/s. Solid lines correspond to CVs measured directly after immersion of the electrode into Pb-containing solution, which are stable with time. Dotted dashed lines correspond to the first CV measured after keeping the potential for periods between 60 and 120 min. at potentials positive of -0.8 V (120 min. at -0.60 V, 60 min. at -0.80 V, for Br-, Cl-containing electrolyte).

The overlayer derives from supersaturationpresumably the system should relax and eventual dissolve or is it the slow or steady supply of Pb from the mercury pool that enables the discrete and stable ultrathin overlayer observed in Fig 3 and b?

Although this topic was not the focus of the present work, we agree that the bulk Pb halide deposits are probably stabilized by the continuous release of Pb, as suggested by the reviewer. This is supported by observations that the deposits are stable for time scales of hours at potentials in the deamalgamation regime, but dissolve rapidly in the regime of amalgamation. This is not the reason

for the formation of the ultrathin “precursor” adlayers, however, which are stabilized by adsorption (see the detailed explanation in the last section of the discussion). We added a corresponding note on the stabilization of the bulk compounds to the discussion.

Corresponding changes in manuscript (p.16):

... the main role of the electrochemical process is to provide a sufficient local Pb^{2+} ion concentration. These ions are continuously released in the dealumination potential regime, leading to a stabilization of the Pb halide bulk deposits.

I was somewhat surprised by the hydroxide component in the adlayer and crystals. What is the solution pH and if the pH is 8 or less shouldn't we expect the kinetics of crystallization to be limited by the available OH^- ?

The formation of $\text{Pb}(\text{OH})\text{Br}$ and $\text{Pb}(\text{OH})\text{Cl}$ in neutral electrolytes was already found in previous studies of the bulk crystal structures, indicating that these compounds are the thermodynamically stable phases. Although we did not study the growth kinetics in detail, the rather slow growth of the bulk deposits may indeed be limited by the concentration of the hydroxide anions, as we now mention in the manuscript. Formation of the precursor adlayer requires only OH^- concentrations in the range of 10^{-9} M and accordingly could occur on time scales of seconds.

Corresponding changes in manuscript (p.11):

The rather slow growth of these deposits may be related to kinetic limitations caused by the low OH^- concentration in the employed neutral electrolytes.

Rather than “quasi-epitaxial” the authors might wish to refer to “highly textured nucleation and growth”.

We coined the term “quasi-epitaxial growth” in our previous publications on PbFBr growth to highlight the template effect of the liquid metal surface. For consistency, we prefer to keep this term, but now define it in the introduction of the manuscript.

Corresponding changes in manuscript (p. 4):

Because of this analogy to solid-on-solid heteroepitaxy, this highly textured nucleation and growth was termed quasi-epitaxial growth, which in the literature is often referred to as fiber texture, was termed quasi-epitaxial growth.

How important is the 0.01 mol/L Na^+ addition to the present work. Did the authors consider its possible role in the adlayer structures.

Na^+ ions are strongly hydrated and non-specifically adsorbing on Hg surfaces. Nevertheless, we considered structures with Na in the crystal structure but no match was found.

For Hg novices the authors may wish to make references to the pzc of Hg and Pb at some point in the paper. The discussion section otherwise is excellent.

We now provide the corresponding potentials of zero charge in the discussion.

Corresponding changes in manuscript (p. 18):

($E_{\text{pzc}} = -0.85 \text{ V}^{56}$, -0.89 V^{59} , and -0.91 V^{56} vs. Hg/HgSO_4 in the F-, Cl-, and Br-containing electrolyte, respectively)

Reviewer 2

This is a very interesting and detailed work on using various techniques, including in situ X-ray scattering studies, to demonstrate that deposition at liquid metal – electrolyte interfaces the chemisorbing ions, such as chloride and bromide, can serve as surfactants that promoting the growth of strongly textured precipitates. I do not have any technical comments on the work, but would ask the authors to comment on how this method can be generalized to other materials (in addition to the toxic ones discussed here) and what concrete role for an upscaled synthesis this approach can have.

We now include a short section on the possible generalization of this method in the conclusions.

Corresponding changes in manuscript (p.20-21):

Furthermore, the same principles should be applicable to the growth of compounds with other cations. For example, Cs and Cu are well known for forming salt-like adlayers with halides on solid electrodes⁶² and thus are likely candidates for similar quasi-epitaxial growth. In addition, these processes might be transferable to other liquid metal substrates⁶⁵, such as InHg alloys²⁶, or less toxic Ga^{29,66,67} which likewise provide atomically smooth substrates and have been already employed in the electrochemical growth of a range of materials. Due to the liquid electrode, upscaling the synthesis for industrial growth is a realistic possibility⁶⁸.